# Development of a Cyclodextrin-Based Mucoadhesive-Thermosensitive In Situ Gel for Clonazepam Intranasal Delivery

**DOI:** 10.3390/pharmaceutics13070969

**Published:** 2021-06-26

**Authors:** Marzia Cirri, Francesca Maestrelli, Giulia Nerli, Natascia Mennini, Mario D’Ambrosio, Cristina Luceri, Paola Angela Mura

**Affiliations:** 1Department of Chemistry, University of Florence, Via Schiff 6, Sesto Fiorentino, 50019 Florence, Italy; marzia.cirri@unifi.it (M.C.); giulia.nerli@unifi.it (G.N.); natascia.mennini@unifi.it (N.M.); paola.mura@unifi.it (P.A.M.); 2NEUROFARBA, Department of Neurosciences, Psychology, Drug Research and Children’s Health, Section of Pharmacology and Toxicology, University of Florence, Viale Pieraccini 6, 50139 Florence, Italy; mario.dambrosio@unifi.it (M.D.); cristina.luceri@unifi.it (C.L.)

**Keywords:** clonazepam, in situ nasal gel, poloxamer, mucoadhesive polymers, randomly methylated β-Cyclodextrin, permeability, cytotoxicity

## Abstract

A thermosensitive, mucoadhesive in-situ gel for clonazepam (CLZ) intranasal delivery was developed, which aimed to achieve prolonged in-situ residence and controlled drug release, overcoming problems associated with its oral or parenteral administration. Poloxamer was selected as a thermosensitive polymer and chitosan glutamate and sodium hyaluronate as mucoadhesive and permeation enhancer. Moreover, randomly methylated β-Cyclodextrin (RAMEB) was used to improve the low drug solubility. A screening DoE was applied for a systematic examination of the effect of varying the formulation components proportions on gelation temperature, gelation time and pH. Drug-loaded gels at different clonazepam-RAMEB concentrations were then prepared and characterized for gelation temperature, gelation time, gel strength, mucoadhesive strength, mucoadhesion time, and drug release properties. All formulations showed suitable gelation temperature (29–30.5 °C) and time (50–65 s), but the one with the highest drug-RAMEB concentration showed the best mucoadhesive strength, longest mucoadhesion time (6 h), and greatest release rate. Therefore, it was selected for cytotoxicity and permeation studies through Caco-2 cells, compared with an analogous formulation without RAMEB and a drug solution. Both gels were significantly more effective than the solution. However, RAMEB was essential not only to promote drug release, but also to reduce drug cytotoxicity and further improve its permeability.

## 1. Introduction

Epilepsy affects more than 50 million people worldwide [1], and benzodiazepines are still considered the mainstay in the treatment of epileptic acute repetitive seizures. Among the different benzodiazepines, clonazepam (CLZ) offers the advantage of longer duration of action [2], proving particularly effective in the treatment of seizures, panic disorder and akathisia [3,4,5]. However, its very low water solubility gives rise to a rate-limited dissolution and variable oral absorption, challenging the development of safe liquid formulations, requiring the use of organic solvents and cosolvents. Moreover, oral or parenteral CLZ administration results in limited uptake across the blood–brain barrier [6] and distribution to no-targeted sites, thus leading to several side-effects, including muscle weakness, drowsiness, dizziness, and unsteadiness [7]. Additionally, other pharmacological issues of this drug, such as high first-pass metabolism, need for long-term treatment, and repetitive dosing, make the development of innovative delivery systems exploiting alternative routes to the oral and parenteral ones particularly desirable.

A very interesting alternative route for an effective CLZ administration can be offered by intranasal delivery. In fact, in recent years there has been an increasing interest in intranasal drug delivery to treat not only local but also systemic diseases, and the nasal route is now considered as a reliable, effective, easily-accessible, non-invasive and painless drug administration way [8,9]. In particular, intranasal delivery was revealed to be an excellent strategy to delivery drugs directly to the central nervous system (CNS), through the olfactory and trigeminal nerve pathways, bypassing the blood–brain barrier (BBB) [10]. The nose-to-brain pathway seems to be especially beneficial for the treatment of different kinds of neurological brain disorders [10,11], and in particular in the treatment of epileptic seizure emergencies by benzodiazepines [12,13]. Commercial intranasal formulations of diazepam and midazolam have been recently approved [13]. Numerous additional benefits, such as avoidance of first-pass metabolism, fast onset of action, reduced side-effects, ease of access and quick self-administration, as well as the possibility to be administered to patients in unconscious state, are associated to this administration route.

The extensive vascularization and the leaky, highly permeable epithelial tissue make the nasal mucosa a favorable drug absorption site [14]. However, the limited surface of nasal cavity, together with the short in situ residence time, and the rapid muco-ciliary clearance (turnover time of 15–20 min) can strongly limit the effectiveness of intranasal drug delivery [15]. Several formulation approaches have been investigated to overcome these hurdles and obtain prolonged residence time and sustained nasal drug release, and, among these, in-situ gel forming systems appear as particularly attractive [16,17]. Among the different environmentally responsive systems investigated, thermosensitive hydrogels with lower critical gelation temperature (LCGT) close to the physiological temperature range are the most used [18,19,20,21]. They have the advantage of being fluid at ambient temperature, thus being easily administrable with high dose accuracy by nasal drops. However, they rapidly undergo to a sol-gel transition at the nasal cavity temperature, thus enabling to both prolong the drug contact time with the mucosa and provide a controlled release.

Poloxamer (PLX) is a thermosensitive non-ionic copolymer of hydrophilic poly (ethylene oxide) and hydrophobic poly (propylene oxide) blocks (PEO-PPO-PEO) and is of great interest in drug delivery, in view of its non-toxicity, very good tolerability, well tunable LCGT and gelation time values, and then wide flexibility and versatility in use [22]. In aqueous solution, in sufficiently concentrated samples, the individual chains of this three-block copolymer reach the critical micellar concentration, changing to self-assembling micelles. In these micelles the hydrophobic PPO segments form the core and the hydrophilic PEO segments the external corona; as temperature increases, PPO block dehydration produces an ordered packing of micelles with different structures, giving rise, at a given critical temperature, and at polymer concentration between 20% and 40%, to the formation of a face-centered cubic structure that produce a closely packed viscous gel [23].

The greater effectiveness of the combined use of in situ gel forming polymers with mucoadhesive polymers in more efficiently prolonging the residence time of the drug delivery system at the administration site, and then in improving drug bioavailability, has been clearly proven [24,25]. Among the numerous polymers investigated for their mucoadhesive properties, chitosan emerged as one of the most widely utilized. In fact, in addition to its mucoadhesive power, related to the interactions between its positively charged amine residues and the negatively charged mucus layer, it offers numerous additional favorable characteristics, including large abundance in nature (being a derivative of chitin), absence of toxicity, high biocompatibility and permeation enhancer properties through the biological membranes [26,27]. Hyaluronic acid (HA), as such or as sodium salt, is another interesting natural polymer particularly suitable as excipient for in situ nasal formulations, considering not only its excellent mucoadhesive capacity, but also its non-toxicity, full biocompatibility, and enhancer properties through mucosal tissues, as well as its ability to restore and maintain mucosal health and functions [28,29].

An adequate drug solubility is a basic requirement for the development of an efficient intranasal delivery system, even more when considering the restricted solution volume which can be accommodated by the nasal cavity (150 µL/nostril). Therefore, due to the very low aqueous solubility of CLZ, the use of cyclodextrins was considered beneficial, in virtue of their proved ability to improve solubility and the relative bioavailability of poorly soluble drugs [30]. The solubilizing effect of different kinds of cyclodextrins towards CLZ has been previously investigated and randomly methylated β-cyclodextrin (RAMEB) emerged as the most efficient one [31]. Moreover, RAMEB showed absorption enhancer properties in nasal drug delivery [32,33] and did not show any toxicity up to 20% concentration [34].

Based on these considerations, the purpose of this work was the development of mucoadhesive, in situ gels for intranasal delivery of CLZ based on PLX as thermoreversible polymer, sodium hyaluronate and chitosan as mucoadhesive and permeation enhancer natural polymers, and RAMEB as drug complexing/solubilizing agent and potential permeation enhancer.

Thermosensitive nasal gels of some benzodiazepines, such as midazolam [35] and lorazepam [36], have been already reported. Instead, to the best of our knowledge, only polymeric micelles [37], mucoadhesive microemulsions [6], or microspheres [38] formulations have been experimented for CLZ intranasal administration. Moreover, although combinations of PLX with chitosan [39], chitosan-cyclodextrin [40], HA [41,42], and chitosan-HA [43] have already been reported for in-situ gel formulations, to the best of our knowledge, this is the first report on the joint use of all these four components for the development of a multifunctional thermosensitive gel.

Statistical design of experiments (DoE) is a useful strategy that can allow a systematic evaluation of the effect of variations of the formulation components proportions, helping to find the optimal composition to reach the prefixed goal with the minimum number of experiments [44]. Therefore, the first step of the study was devoted to investigating, by a DoE methodology, the effect of varying the respective amounts of the formulation components on gelation time, gelation temperature, and pH, considered as the critical responses to be optimized for the development of an efficient, thermosensitive, in situ nasal gel. Once defined the best formulation composition, drug-loaded batches were prepared and further characterized for rheological, mucoadhesive and in vitro release properties. The final selected formulation was also evaluated for cytotoxicity and drug permeation using Caco2 cells.

## 2. Materials and Methods

### 2.1. Materials

Clonazepam (5-(2-chlorophenyl)-7-nitro-3H-1,4-benzodiazepin-2(1H)-one) (CLZ) and Poloxamer 407 (PLX) were purchased from Sigma (St. Louis, MO, USA). Randomly methylated β-cyclodextrin (RAMEB) was kindly donated by Wacker-Chemie Italia SpA, (Milan, Italy). Chitosan glutamate (Chito Gl) (Protosan G213) was purchased from Novamatrix (Sandivka, Norway) and Hyaluronic Acid sodium salt (HA) (Hyacare^®^) was kindly donated by Novozymes Biopharma US Inc. (Cambridge, MA, USA). All other chemicals and solvents were of analytical grade.

### 2.2. HPLC Assay of Clonazepam (CLZ)

CLZ assay was performed by HPLC (Merck Hitachi Elite LaChrom apparatus, Darmstadt, Germany) endowed with a L-2400 UV-vis detector and a L-2130 isocratic pump. An acetonitrile/water 30:70 *v*/*v* mixture was used as mobile phase. An Agilent Zorbax CN column (150 × 4.6 mm, 5 µm particle size) was the stationary phase. UV detection was performed at 310 nm. The injection volume 20 µL, the flow rate 0.9 mL/min, the column temperature 40 ± 1.0 °C. Under these conditions, the CLZ retention time was 7.23 ± 0.01 min. The method was validated for linearity (r^2^ = 0.9985), limit of quantification (0.27 µg/mL) and limit of detection (0.08 µg/mL).

### 2.3. Experimental Design

The MODDE software (MODDEGo version 12.01, Umetrics, Malmö, Sweden) was used for the statistical experimental design and the analysis of the obtained results. Relations between dependent and independent variables were evaluated by multiple linear regression analysis (MLRA). Significance and validity of the model was tested by analysis of variance (ANOVA).

### 2.4. Thermosensitive Gels Preparation

Thermosensitive gels were prepared according to the “cold method” [40]. Briefly, PLX was dissolved under stirring in 10 mL of deionized water at room temperature, then cooled down at 4 °C and kept overnight in a refrigerator to ensure complete polymer hydration and dissolution. The day after, the other components were added in sequence, under continuous stirring at 4 °C up to their complete dissolution. The preparation method was the same for drug-loaded gel formulations, where CLZ was added to the solution after RAMEB addition, under stirring until complete dissolution. The gels were kept at 4 °C before use.

### 2.5. Characterization of Thermosensitive Gels

#### 2.5.1. Determination of pH

The pH was measured using a Crison Basic 20 pH-meter (Crison Instruments, Alella, Barcelona, Spain). The results are the mean of 3 separate measurements.

#### 2.5.2. Determination of Gelation Temperature

The gelation temperature was measured by the “vial inverting” method [42]. A 10 mL sample was poured in a vial and put in a water bath at 15 °C, whose temperature was increased by 2 °C every 5 min, up to 25 °C, and then by 1 °C every 5 min. The sol-gel transition temperature was determined by inverting the vials at every step. The temperature at which the formulation did not flow, even when inverted, was recorded as gelation temperature. Results are the mean of 3 measurements.

#### 2.5.3. Determination of Gelation Time

The gelation time was determined according to a previous work [45]. Briefly, a vial containing 10 mL of formulation sample and a magnetic bar was immersed in a thermostated bath at 34 °C, considered as the nasal physiological temperature, under continuous stirring at 200 rpm. The time at which the magnetic bar stopped shaking, as a consequence of gelation, was recorded as the gelation time. The results are the mean of 3 separate determinations.

#### 2.5.4. Determination of Gel Strength

The gel strength was evaluated by the method described by Yong et al. [46] opportunely modified. Briefly, a formulation sample (20 mL) was put into a graduated 50-mL graduated cylinder and allowed to gel in a thermostated bath at 34 °C. A weight of 15 g was then put on the gelled solution. The gel strength, index of gel viscosity at the nasal physiological temperature, was measured as the time necessary to the weight to penetrate 1 cm into the gel. The results are the average of 3 separate measurements.

#### 2.5.5. Determination of Gel Mucoadhesive Strength

Mucoadhesive properties were evaluated by a modified two-arms balance method [47]. A gel of porcine mucine (2%) agar (1.5%), used as a model of mucoadhesive tissue, was put in a dish (2.5 cm diameter). A constant amount of previously gelled formulation was interposed between such dish and another empty dish. The first dish was attached to one plate of the balance, while the other one was connected to a mobile support, which was carefully lifted up to counterbalance the weight of a container placed on the other balance plate. After a pre-set contact time (5 min), water was added into the container at a constant rate of 10 mL/min by a peristaltic pump. The water amount needed to obtain the detachment of the two surfaces, expressed in g/cm^2^, was recorded as mucoadhesive strength. The results are the mean of 3 separate measurements.

#### 2.5.6. Determination of Mucoadhesion Time

Mucoadhesive power in terms of in situ residence time of the gel formulation was evaluated by following the method of Khan et al. [48], slightly modified. Briefly, a formulation sample (1 g), added with 0.1% of carminic acid, was left to gel in a thermostated water bath at 34 °C and then applied on a Petri dish containing a porcine mucine (2%) agar (1.5%) gel as mucoadhesive tissue model. This set was put at an angle of 45° in a thermostated chamber (34 °C) and subjected, by a peristaltic pump, to a 1 mL/min flux of a pH 6.5 simulated nasal fluid (SNF) solution [36] mimicking the physiological conditions at the nasal cavity. The time needed for full formulation washing, established on the basis of the color disappearance, was measured as the mucoadhesion time. The test was repeated 3 times for each formulation and the results averaged.

#### 2.5.7. Rheological Studies

The rheological behavior of gel formulation was determined at 37 °C using a rotational Brookfield viscometer (RV Model, Brookfield, WI, USA) by the continuous shear method. The ascending and descending flow curves were recorded. The experiments were repeated three times and the results averaged.

### 2.6. In Vitro Drug Release Studies

The dialysis technique was applied for in vitro release studies [42]. Briefly, two mL of gel were introduced in a cellulose acetate dialysis bag (MWcut-off 14 kDa, Sigma-Aldrich, St. Louis, MO, USA) previously soaked in SNF and then placed in 15 mL of SNF as release medium. The systems were kept at 34 °C, and magnetically stirred at 100 rpm. At predetermined time intervals (1, 2, 3, 4, 5, 6 h), 0.5 mL of medium was withdrawn and quickly replaced with the same volume of fresh medium. A correction for the cumulative dilution was made. All the experiments were performed in triplicate. The amount of CLZ released was assayed by HPLC, as described above (Section 2.2). Drug release data were fitted into zero order, first order, Higuchi, and Korsmeyer–Peppas mathematical models to identify the best fitting kinetic model. The drug release mechanism was also inferred by the value of the exponent “*n*” of the Korsmeyer–Peppas equation:*Mt/M∞* = *K × t^n^*(1)
where *Mt*/*M∞* is the fraction of drug released at time *t*, *K* the release constant, and *n* the release exponent.

### 2.7. Cytotoxicity and Transport Studies

Caco-2 (human colorectal adenocarcinoma) cell line coming from the American Type Culture Collection (ATCC, Rockville, MD, USA) was cultured in Dulbecco’s modified Eagle’s medium (DMEM, Euroclone, Milan, Italy), added with 20% fetal bovine serum (FBS), 1% L-glutamine and 1% penicillin-streptomycin (Carlo Erba, Milan, Italy). The cells were maintained at 37 °C in a humidified incubator in an atmosphere containing 5% CO_2_.

Viability analyses were performed by a CellTiter 96^®^ Aqueous One Solution Cell proliferation (3-(4,5-dimethylthiazol-2-yl)5-(3-carboxymethoxyphenyl)-2-(4-sulfophenyl)-2H-tetrazolium) inner salt (MTS) assay kit (Promega Corporation, Madison, WI, USA). This is a colorimetric method based on MTS reduction by viable cells in a colored product (formazan). Briefly, Caco-2 cells were seeded in 96-well culture plates at a density of 5 × 10^3^ cells/well. At approximately 80% of confluency, cells were incubated for 2 h with different dilutions (1:10, 1:50, 1:100) of the gel formulations and of a CLZ solution in dimethyl sulfoxide (DMSO) at the same drug concentration. The optical density of the formed chromogenic product was measured at 490 nm with a Multilabel Counter (Victor3 Wallac 1240, Perkin Elmer, Waltham, MA, USA). The relative cell viability was expressed as a percentage of viable cells compared to the untreated control group.

For transport studies, cells were seeded at 2 × 10^5^ cells/well and allowed to grow and differentiate for 21 days. Culture medium was added to apical (AP) and basolateral (BL) side and replaced every second day for the first week and daily thereafter. The cell layer integrity was checked by the Lucifer Yellow (LY) permeability assay. LY was diluted in the transport buffer (Hank’s Balanced Salt Solution (HBSS/HEPES, 1M)) and added to the AP compartment at a final concentration of 100 µM. After 1 h incubation at 37 °C, the HBSS in the BL chamber was collected, and LY concentration determined by a fluorescence plate reader (Victor3 Wallac 1240, Perkin Elmer, Waltham, MA, USA) at 485 nm excitation and 530 nm emission. The AP-to-BL% permeability was calculated by this equation:% Permeability = (Fluorescence in BL-blank)/(Fluorescence in AP-blank) × 100(2)

The critical maximum LY flux, index of leaky monolayers, was fixed at 5% of the initial amount.

Transport studies were carried out as previously reported [49]. Briefly, the culture medium was replaced by the HBSS/HEPES transport buffer, previously heated at 37 °C. Caco-2 cells were then exposed for 2 h with gel formulations or CLZ solution at the same drug concentration, suitably diluted, in the AP chamber. At given time intervals, 0.3 mL of medium was withdrawn from the BL side for CLZ assay by HPLC and replaced with the same volume of fresh HBSS/HEPES. At the end of the experiment, the cell layer integrity was rechecked by the LY permeability assay, as described above. The apparent permeability coefficient (P_app_) was calculated according to this equation [50]:(3)$$P_{\mathrm{app}}=\frac{{dC}_{BL}}{dt}\times \frac{1}{{A\;\times \;C}_{AP}}$$
where *C_BL_* is the CLZ concentration (mg/mL) in the basolateral (*BL*) (acceptor) compartment as a function of time *t* (s), *A* the surface area of the Transwell membrane (1.13 cm^2^), and *C_AP_* the initial CLZ concentration in the *AP* (donor) chamber (mg/mL).

### 2.8. Statistical Analysis

Statistical analysis of data was carried out by one-way analysis of variance (ANOVA), testing differences between groups by Student–Newman–Keuls comparison post-hoc test (GraphPad Prism version 6.0 Software, San Diego, CA, USA). *p*-values < 0.05 or <0.01 were considered significant.

## 3. Results

### 3.1. Pre-Formulation Studies with Design of Experiment (DoE)

A suitable gelation temperature, lower than 34 °C, considering the physiological nasal temperature [18], and proper gelation time, which should be around 60 s, as a good compromise allowing, on one hand, an easy drug administration and, on the other, to prevent dripping from the nasal cavity and avoid rapid nasal clearance [51], are critical parameters in the development of an effective in situ thermosensitive nasal gel formulation. Another important factor to be considered for the development of a well tolerable and not-irritant nasal formulation is its pH, which should be in the range of the physiological pH of nasal mucosa, i.e., between 5.5 and 6.5 [52].

It is known that PLX gelation temperature increases with decreasing its concentration, and that combinations of PLX with other excipients may alter the dehydration of hydrophobic PPO blocks and the consequent micellization process, thus changing its sol-gel transition temperature and time necessary for gelling [23]. Moreover, ionizable excipients can affect the formulation pH.

A screening design was therefore carried out as a first step of this work, in order to investigate the effect of varying the relative percent of the gel formulation components on gelation temperature (Y_1_), gelation time (Y_2_), and pH (Y_3_), considered as the responses (dependent variables) to be optimized, as well as to point out possible interactions among the components. A 2^3^ (three factors-two-levels) full factorial design was applied. The amounts of PLX (X_1_), HA (X_2_), and RAMEB (X_3_) were selected as the independent variables (factors). The Chito Gl concentration was instead kept constant at 0.1% *w*/*v*, since preliminary experiments showed that higher amount of this polymer (0.2% or 0.3%) in the gel formulation gave rise to a progressive reduction of drug release rate. Similar results, regardless of the chitosan molecular weight, were observed by other authors for a chitosan-PLX in situ ocular gel [53].

The experimental domain of each factor, determined on the basis of subject-matter knowledge and previous studies [23,31], was set as follows: PLX (X_1_): 20–24% *w*/*v*; HA (X_2_) 0.0–0.5% *w*/*v*; CD RAMEB(X_3_): 3.0–5.0% *w*/*v*.

The minimum, target and maximum values for gelation temperature (28, 30, 32 °C) and pH (5.5, 6.0, 6.5) responses were defined on the basis of the physiological conditions at the nasal mucosa. As for gelation time, a target value of 60 s, with a range of ±30 s, was selected, considering it suitable for enabling both an easy administration and a sufficiently fast gelling process [51]. The relations between dependent and independent variables were evaluated by MLRA, according to the following mathematical model:*Y* = *β*_0_ + *β*_1_*X*_1_ + *β*_2_*X*_2_ + *β*_3_*X*_3_ + *β*_12_*X*_1_*X*_2_ + *β*_13_ *X*_1_*X*_3_ + *β*_23_ *X*_2_*X*_3_ + *β*_123_ *X*_1_*X*_2_*X*_3_(4)
where *Y* is the dependent variable, while *β*_0_ is a constant, *β*_1_ to *β*_3_ are the coefficient computed from the observed experimental values of the independent variables, and the terms *X*_1_*X*_2_ to *X*_1_*X*_2_*X*_3_ are the interactions among the factors.

The experimental plan of 11 experiments (including the 3 centre points) is shown in Table 1. Formulations were prepared according to this experimental plan and evaluated for gelation temperature, gelation time, and pH.

ANOVA analysis of the experimental results indicated that the assumed regression model was statistically significant (*p* < 0.05) and valid for all the responses investigated.

The influence of varying the concentration of each factor on the three considered responses was evaluated by the summary of fit plots of four parameters: R^2^ (% of variation of the response explained by the model), Q^2^ (% of variation of the response predicted by the model according to cross validation, expressed in the same units as R^2^), model validity and reproducibility (variation of the replicates compared to overall variability).

The summary of fit plots (Figure 1) gives information about strength and robustness of the model: R^2^ should be as closer as possible to the ideal value 1 and Q^2^ should be ideally >0.5 for a good fit and predictive power. Values less than 0.25 for model validity indicate statistically significant model problems. Reproducibility values should be greater than 0.5.

As can be seen in Figure 1A, R^2^ ≥ 0.92 were obtained in all cases, indicating a strong fit between data and model. Q^2^ was 0.85 and 0.80 for gelation temperature and gelation time, respectively, suggesting a very high prediction precision. It resulted instead somewhat lower for pH, though still being clearly higher than 0.5, then indicating a satisfying predictive power also for this response. High model validity values were obtained for gelation temperature and pH responses (0.77 and 0.54, respectively), while a lower value was observed for gelation time, but still considerably higher than the threshold value of 0.25, denoting the absence of statistically significant model problems also for this response. The high values obtained in all cases for reproducibility (≥0.96), all much higher than the requested value of 0.5, indicated good experimental control and low pure error.

The goodness of the model was also confirmed from the observed versus predicted responses plots (Figure 1B). Almost all the experimental values were very close to the predicted ones, meaning that the model was adequate and had a good predictive power. Moreover, the R^2^ values very near to the ideal value 1 indicated that the regression model fitted the raw data very well.

Scaled and centred coefficients plots were applied to investigate the individual contribution of each factor to the evaluated responses (Figure 2). The factor could be considered a significant parameter affecting the response when its confidence interval does not cross the horizontal axis.

PLX %, as expected, was the main factor affecting the gelation temperature, that tends to decrease with increasing PLX concentration. An opposite effect was instead found for the PLX*RAMEB interaction term, indicating an increase in gelation temperature as a consequence of interactions between RAMEB and thermosensitive polymer. Even in the case of the gelation time, PLX % (as such and as quadratic term) was the main factor influencing the response: an increase in PLX % resulted in a decrease of the gelation time. Moreover, the response was also affected by HA %, showing a decrease with increasing the polymer content. An opposite effect was instead observed in the case of RAMEB %, whose increase gave rise to an increase of gelation time, even though of borderline statistical significance. Finally, pH was significantly affected only by HA % variations.

Response 4D contour plots were created to obtain an overall view of the changing tendency of the responses and find the optimal values of the factors to obtain the desired responses (Figure 3).

As can be seen in the 4D contour plot for the gelation temperature, the PLX % needed to achieve the target value of this response (30 °C) was near to 22% (precisely 21.8%). Moreover, as indicated by its almost vertical line, the PLX % corresponding to the target value of this response was independent not only, as expected, from HA %, but also from the RAMEB %. In fact, the above observed incremental effect of RAMEB % on the gelation temperature (see Figure 2) became evident only at lower or higher PLX %.

Regarding the gelation time, the target value of 60 s was obtained with different combinations of the three considered variables. In particular, for 0.25% HA (corresponding to the target pH value), the PLX % to reach the target gelation time was near to 23%, and slightly increased with increasing the RAMEB content. However, it was considered more opportune to give priority to the achievement of gelation temperature target value, and then to maintain 22% as PLX content, considering gelation times that ranged around 70 s (obtained for such polymer concentration while varying the RAMEB content from 3% to 5%), still acceptable

Finally, the target pH value of 6.0 was instead achieved for HA contents between 0.2 and 0.27% *w*/*v*. In particular, at 0.25% HA, the target pH was almost unaffected by PLX and RAMEB content.

### 3.2. Preparation and Characterization of Drug-Loaded Gel Formulations

The DoE screening design indicated PLX and HA content as the main factors affecting the considered gel properties, and also suggested their optimal concentrations to obtain the desired responses. On the basis of DoE results, drug-loaded gel formulations were prepared using the optimized amounts of PLX (22%) and HA (0.25%) (Table 2). The composition of the examined gel formulations is reported in Table 2.

As can be seen, in the case of RAMEB, the third “factor” considered in the screening phase, owing to its important role in improving CLZ solubility and, consequently, its release properties, it was considered appropriate to continue to test the effect of the three concentration levels, which allowed solubilization of different CLZ amounts (Table 2). Analogous formulations not containing the drug were also prepared (G2, G4 and G6), with the purpose of evaluating any possible effect of the CLZ addition on the gel physicochemical properties. Moreover, reference formulations containing PLX at 22% alone (G0) or in the presence of HA 0.25% (G0A) or Chito Gl 0.1% (G0B), or both (G0AB), were also prepared (Table 2).

All the formulations were evaluated in terms of gelation temperature, gelation time, gel strength, and mucoadhesive strength. The results are summarized in Figure 4.

The gelation temperature of all the batches was in the range between 28.5 and 30.5 °C, considered suitable for in situ thermosensitive nasal gel formulations. It was observed that the presence of RAMEB led to a slight increase in gelation temperature with respect to the formulations without it. This phenomenon, that agreed with the DoE results, was observed by other authors for hydroxypropyl-β-cyclodextrin and ascribed to possible hydrogen bond formation between the cyclodextrin hydroxyl groups and the propylene oxide chains of PLX micelles, which hindered micelle–micelle interactions, necessitating a higher temperature for polymer dehydration, with a consequent reduction of the gelling capacity of the thermosensitive polymer [54]. An analogous effect was reported by other Authors for dimethyl-β-cyclodextrin, and attributed to interactions between the methyl groups of this cyclodextrin derivative and PLX, which reduced the activity of PPO segments, thus increasing the critical gelation temperature [55].

Gelation time was in the right range between 45–65 s for all the formulations. As in the case of gelation temperature case, no effects of the drug presence were detected. The slight increase in gelation time observed for formulations containing RAMEB (also this in agreement with DoE results) was considered related to the concomitant increase of gelation temperature, and attributed to interactions between RAMEB and PLX that delayed the formation of the three-dimensional structure of the gel [54,55].

The gel strength of drug-loaded formulations (G1, G3, and G5) was not significantly different from that of the corresponding formulations without drug (G2, G4, and G6), and also similar to that of PLX alone (G0) and the reference formulation not containing RAMEB and CLZ (G0AB).

Interestingly, the reference formulation G0AB showed the highest mucoadhesive strength, about 1.4 times higher than that of PLX alone (G0), pointing out the favourable effect of the combined use of the two mucoadhesive polymers, HA and Chito Gl. On the other hand, the addition of RAMEB to the formulation (irrespective of the presence or not of CLZ) gave rise to a slight reduction of mucoadhesive strength which, however, always remained significantly higher than that of PLX alone. This effect was less evident for the formulation with the highest RAMEB concentration (G5), whose mucoadhesion strength was similar to that of G0AB.

The mucoadhesion time, i.e., the in-situ residence time at level of the nasal mucosa, is another important parameter that have to be taken into consideration, since it is essential for assuring an intimate contact of the formulation at local level, and a prolonged release of the drug. All the drug-loaded batches exhibited satisfactory in vitro mucoadhesion times, ranging from 4 to 6 h. Interestingly the longest mucoadhesion time (6 h) was obtained for the formulation with the highest drug-RAMEB content (G5), which also showed the greatest mucoadhesion strength.

Rheological studies indicated that all gel formulations showed analogous pseudo-plastic flow behaviour without thixotropy, characterized by a drastic viscosity decrease as soon as the shearing began. Shear tinning properties are considered desirable for thermosensitive hydrogel aimed for nasal administration [56]. The drug-loaded gel formulations exhibited almost superimposable rheograms, thus excluding possible differences in drug release rate related to the different gel viscosity.

### 3.3. In Vitro Drug Release Studies

In vitro drug releases profiles from G1, G3 and G5 formulations are shown in Figure 5. In order to evaluate the possible effect of RAMEB on the drug release rate, they were compared to the drug release curves obtained from the corresponding formulations not containing RAMEB. As expected, the amount of drug released as a function of time increased with increasing the CLZ content in the formulation. However, interestingly, a clear favorable effect of RAMEB in improving the drug release rate was observed. In fact, the amount of drug released at each time from G1, G3, and G5 formulations was always significantly higher than that from the corresponding formulations without RAMEB. This result could be reasonably attributed to the solubilizing effect of RAMEB towards CLZ [31].

Drug release data were modeled according to different mathematical models, i.e., zero and first order kinetics, Higuchi and Korsmeyer–Peppas equations, in order to identify the best fitting kinetic model on the basis of the values of the corresponding correlation coefficients (R^2^). For all the gel formulations, CLZ release was better described by the Korsmeyer–Peppas model, as indicated by the highest R^2^ values, ranged between 0.996–0.998. The release exponent “n” of the Korsmeyer–Peppas equation, indicative of the drug release mechanism, was typical of a non-Fickian (anomalous) transport, being ranged between 0.5788–0.6738. This result seems to suggest that both drug diffusion and polymer swelling processes were involved in determining the drug release from the gel formulations.

Considering the results provided by the characterization and release studies, the batch G5 was finally selected for further investigations, having proved to be the formulation endowed with the highest mucoadhesive strength and longest mucoadhesion time, together with suitable values of both gelation temperature and gelation time, and the greatest drug release rate.

### 3.4. Cytotoxicity and Transport Studies through Caco-2 Cells

MTS assay was performed to assess the possible cytotoxicity of the selected gel formulation G5. A CLZ solution at the same concentration as in G5, as well as a gel formulation with the same composition of G5, but without RAMEB, were also tested for comparison purpose. Test samples were prepared by diluting (1:10, 1:50, 1:100) with HBSS/HEPES the starting samples. As shown in Figure 6, cells viability % after 2 h exposure of Caco-2 cells at the 1:50 and 1:100 dilutions was in all cases very close to that of the untreated control cells (100%), indicating total absence of cytotoxicity. Interestingly, cells exposed to the selected formulation (G5) showed a still good viability, around 80%, also at the highest tested concentration (1:10 dilution), while it dropped below 70% for the gel formulation without RAMEB and at about 55% for the CLZ solution. The results seem to indicate a reduction of CLZ cytotoxicity when formulated as gel, particularly marked in the presence of RAMEB, probably in virtue of its complexing power towards the drug.

### 3.5. Transport Experiments with Caco-2 Cells

In order to explore the release behavior of CLZ from the selected in situ nasal gel formulation G5, cellular permeation experiments on Caco-2 cells were carried out. As in the previous cytotoxicity study, a gel formulation analogous to G5, but without RAMEB, and a simple CLZ solution were also tested, in order to evaluate the effect of the gel formulation, and or the presence or not of the RAMEB on the drug permeation properties. The dilution 1:50 was selected for transport studies. Lucifer Yellow (LY) concentrations tested in the basolateral (BL) compartment at the end of the experiments were always lower than 3%, indicating the maintenance of integrity of the cell layer, that confirmed the good tolerability of the tested formulations.

As can be seen in Figure 7, both gel formulations gave rise to a significant increase (*p* < 0.01) in CLZ apparent permeability (P_app_) with respect to the drug solution. Interestingly, the CLZ P_app_ from formulation G5 was significantly higher (*p* < 0.05) than that from the corresponding formulation not containing RAMEB. These results not only confirmed the known permeation enhancer ability of Chito Gl, mainly attributed to its ability to interact with negatively charged glycoproteins on the mucosa surface and transiently open tight junction [26], but also further highlighted the favorable effect of the RAMEB presence in the gel formulation.

In fact, it not only allowed to improve release rate and decrease cytotoxicity of CLZ, in virtue of its solubilizing and complexing power towards the drug, but also exhibited a permeation enhancer effect that added up to those of Chito Gl.

The ability of cyclodextrins, and in particular methylated-β-cyclodextrin, in enhancing the nasal absorption of drugs has already been reported [32,34] and mainly attributed to their capacity to reversibly remove phospholipids and cholesterol from the membrane’s external bilayer, thus temporarily increasing their permeability without toxic effects. A possible transient opening of the tight junctions, as a consequence of such interactions, has also been speculated [57].

## 4. Conclusions

In the present study, a cyclodextrin-based thermosensitive mucoadhesive in situ gel for the intranasal systemic delivery of CLZ, which aimed to provide prolonged in-situ residence and a controlled drug release, was successfully developed.

The formulation components were PLX, as thermosensitive polymer, Chito Gl and HA as mucoadhesive, biocompatible, and permeation enhancer polymers, and RAMEB, as the most effective cyclodextrin for improving the very low drug solubility [31].

A screening DoE was applied in the first step of the work to systematically examine the effect of changes in the proportions of the formulation components, on gelation temperature, gelation time, and pH, considered as the critical responses to be optimized.

Then, based on the results of these preliminary studies, a series of drug-loaded gel formulations, containing three different CLZ-RAMEB concentrations, were prepared and characterized in depth, also considering other important formulation parameters such as gel strength, mucoadhesive strength, mucoadhesion time, and drug release properties.

All formulations showed properties suitable for an in situ mucoadhesive thermosensitive gel formulation, but the one with the greatest drug-RAMEB concentration emerged as the best, particularly in terms of higher mucoadhesion properties, longer mucoadhesion time, and greater drug release rate. Therefore, it was finally selected for cytotoxicity and permeation studies through the Caco-2 cell monolayer, in comparison with a corresponding formulation without RAMEB and a simple drug solution.

Both gel formulations were significantly more effective than the drug solution in improving CLZ permeation. However, the presence of RAMEB proved to be essential not only to promote drug release, as a consequence of the improved drug solubility, but also to reduce drug cytotoxicity, maybe due to the inclusion complex formation, and further improve its permeability, probably in virtue of the RAMEB permeation enhancer effect.

## Figures and Tables

**Figure 1 pharmaceutics-13-00969-f001:**
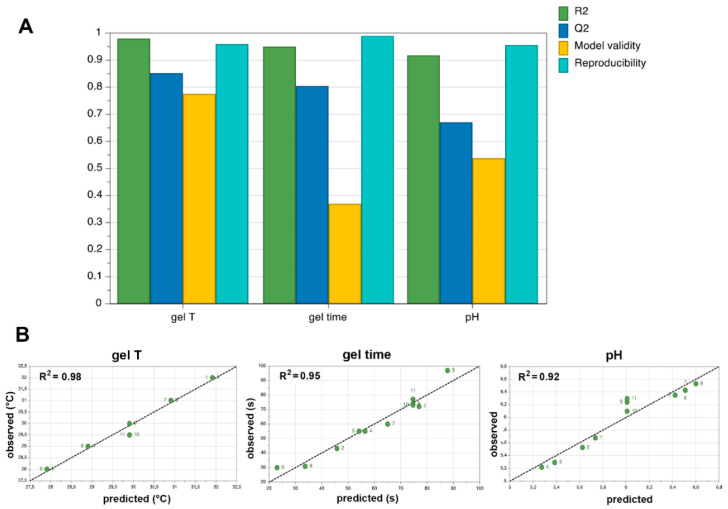
(**A**) Summary of fit plots showing model fit (R^2^), predictability (Q^2^), Model Validity and Reproducibility; (**B**) Observed vs predicted plots of gelation temperature, gelation time and pH.

**Figure 2 pharmaceutics-13-00969-f002:**
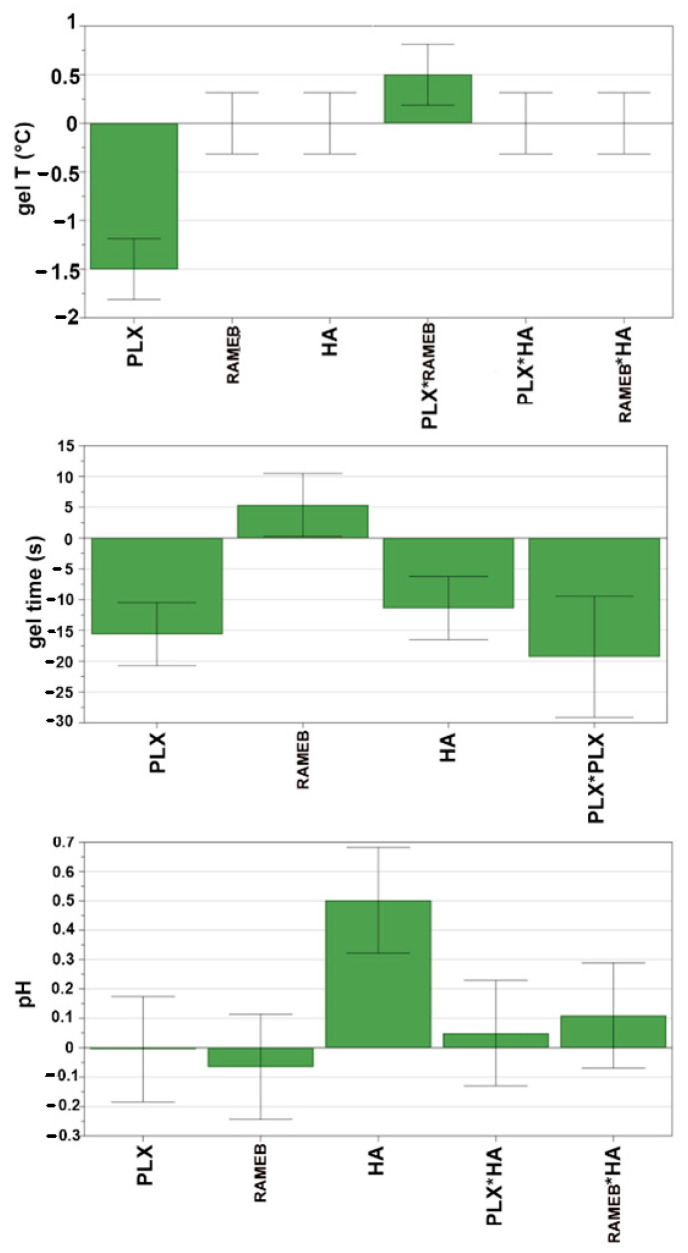
Coefficients plots for gelation temperature, gelation time and pH. Positive and negative bars indicate, respectively, an increment or decrement of the response caused by the term. * is the interaction between the two terms. Non-significant terms have error bars that cross the horizontal axis.

**Figure 3 pharmaceutics-13-00969-f003:**
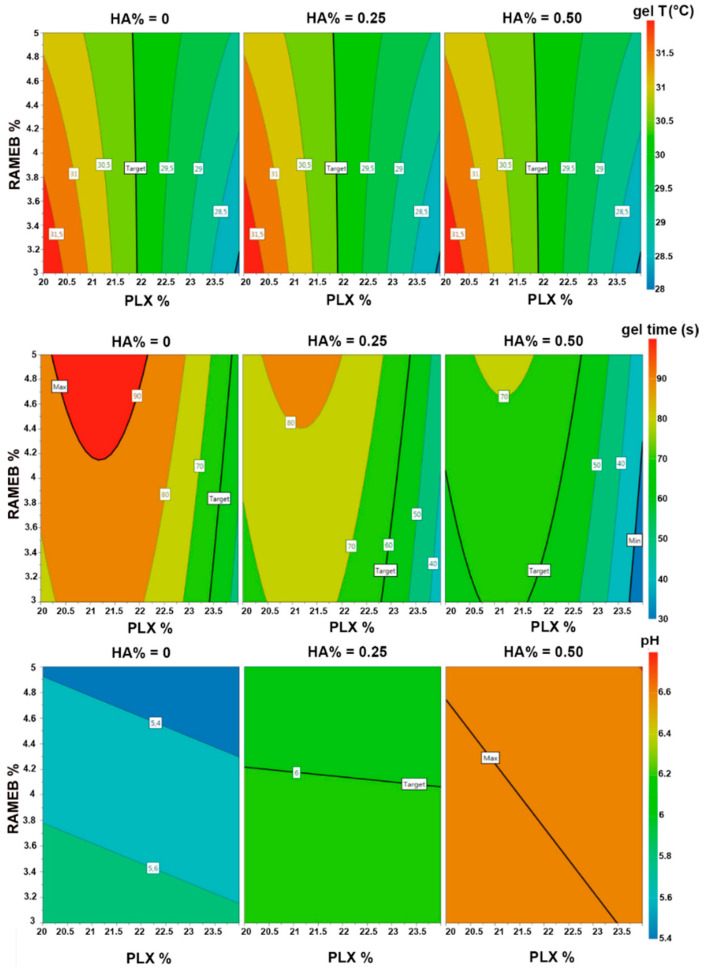
Responses 4D contour plots of gelation temperature, gelation time and pH.

**Figure 4 pharmaceutics-13-00969-f004:**
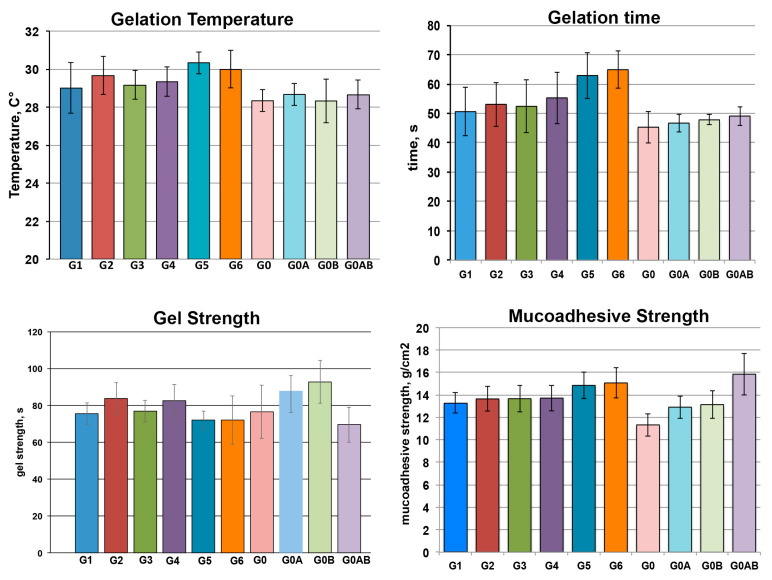
Gelation temperature, gelation time, gel strength and mucoadhesive strength of various thermosensitive mucoadhesive gel formulations (see Table 2 for gel compositions). Data are mean values ± standard deviation (s.d.) (*n* = 3).

**Figure 5 pharmaceutics-13-00969-f005:**
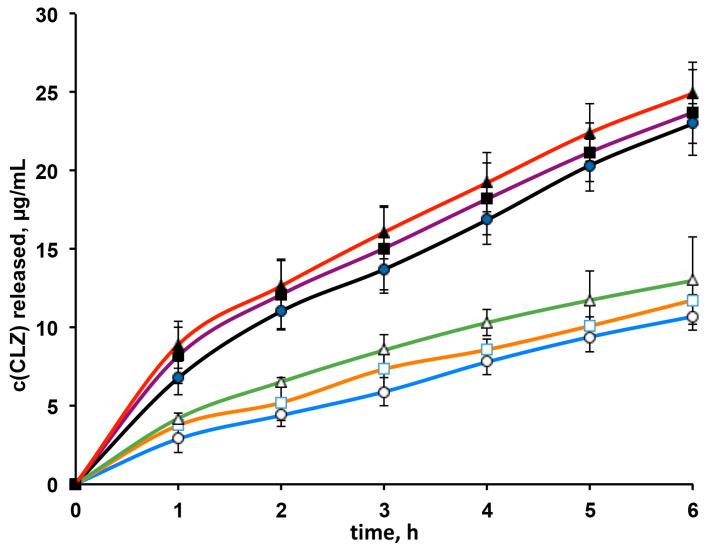
Release profiles of clonazepam (CLZ) from G1 (●), G3 (■) and G5 (▲) gel formulations and from the corresponding ones without RAMEB (white symbols).

**Figure 6 pharmaceutics-13-00969-f006:**
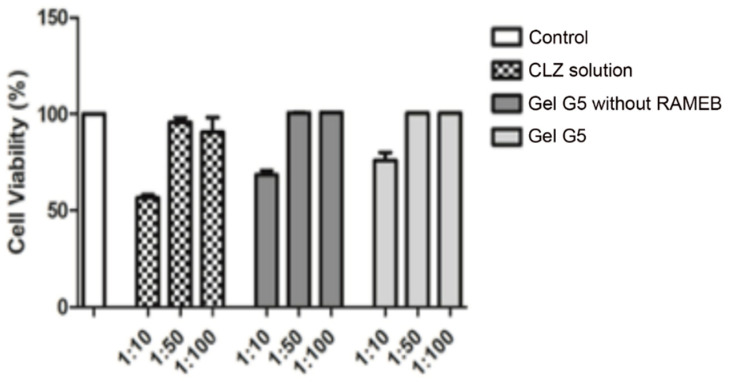
Caco-2 cells viability after 2 h exposure to the clonazepam (CLZ) gel formulation G5 (see Table 2 for composition), or the corresponding formulation without RAMEB, or a CLZ solution, all at different dilutions (1:10, 1:50 and 1:100).

**Figure 7 pharmaceutics-13-00969-f007:**
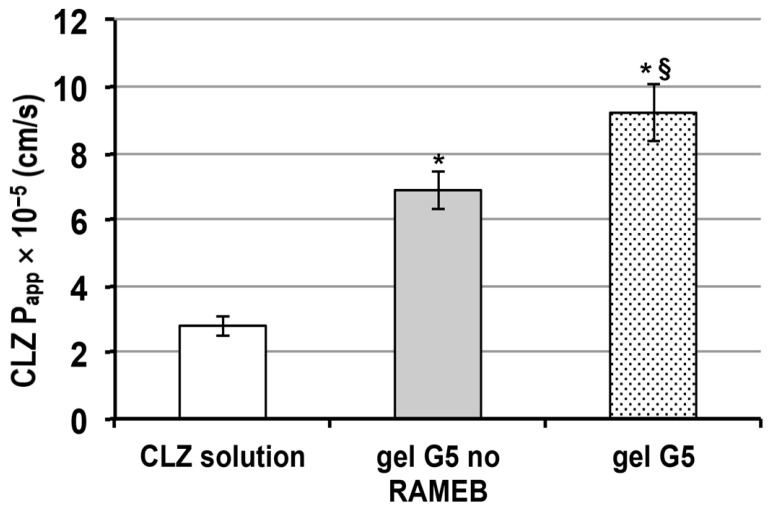
Apparent permeability coefficient (P_app_) of clonazepam (CLZ) as solution, or as gel G5 formulation, in the absence or in the presence of RAMEB (see Table 2 for gel composition). * *p* < 0.01 vs. CLZ solution; § *p* < 0.05 vs G5 no RAMEB.

**Table 1 pharmaceutics-13-00969-t001:** Experimental plan and obtained responses (gelation temperature, gelation time, pH). Variables: PLX %, HA %, RAMEB % (fixed component: Chito Gl 0.1%).

Exp. No	PLX % *w*/*v*	HA % *w*/*v*	RAMEB % *w*/*v*	Chito Gl % *w*/*v*	Gel. T °C	Gel Time s	pH
1	20	0	3	0.1	32.0	72	5.68
2	24	0	3	0.1	28.0	43	5.53
3	20	0	5	0.1	31.0	97	5.29
4	24	0	5	0.1	29.0	55	5.22
5	20	0.50	3	0.1	32.0	55	6.35
6	24	0.50	3	0.1	28.0	30	6.43
7	20	0.50	5	0.1	31.0	60	6.43
8	24	0.50	5	0.1	29.0	31	6.53
9	22	0.25	4	0.1	30.0	74	6.24
10	22	0.25	4	0.1	29.5	73	6.10
11	22	0.25	4	0.1	29.5	77	6.30

**Table 2 pharmaceutics-13-00969-t002:** Composition of the gel formulations.

Batch Code	PLX (% *w*/*v*)	HA (% *w*/*v)*	Chito Gl (% *w*/*v)*	RAMEB (% *w*/*v*)	CLZ (% *w*/*v*)
G1	22	0.25	0.1	3	0.060
G2	22	0.25	0.1	3	/
G3	22	0.25	0.1	4	0.080
G4	22	0.25	0.1	4	/
G5	22	0.25	0.1	5	0.095
G6	22	0.25	0.1	5	/
G0	22	/	/	/	/
G0A	22	0.25	/	/	/
G0B	22	/	0.1	/	/
G0AB	22	0.25	0.1	/	/

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
