# Peer review of "Development of a Cyclodextrin-Based Mucoadhesive-Thermosensitive In Situ Gel for Clonazepam Intranasal Delivery"

_pharmaceutics, 2021, doi:10.3390/pharmaceutics13070969_

Round 1
Reviewer 1 Report
Manuscript ID: Pharmaceutics-1258682
Brief summary:
Undoubtedly, development of novel multicomponent gels as drug delivery systems capable of achieving prolonged residence time and controlled drug release is a research field of high interest in pharmacy. Application of drug-loaded gels eases intranasal administration that is currently considered as a reliable, effective, easily-accessible, non-invasive and painless drug administration way. Therefore, the manuscript fully complies with the scope of the Pharmaceutics journal.
Broad comments
By using a combination of a thermoadhesive polymer, permeation enhacer and mucoadhesive component together with a cyclodextrin derivative as a drug solubilizer, authors have developed a highly-efficient intranasal drug delivery gel. Furthermore, authors have applied the Statistical design of experiments (DoE) considering few critical variables (pH, gelation time, gelation temperature) in order to obtain an optimal formulation of their multicomponent drug-loaded gel (Poloxamer, as thermosensitive polymer, Chitosan Gl and Hyaluronic acid as mucoadhesive, biocompatible and permeation enhancer polymers, and RAMEB cyclodextrin derivative as drug solubility promotor) with a minimum number of trials. Of course, use of this statistical methodology has already been applied in previous articles (i.e.: Hagedorn et al. International Journal of Pharmaceutics (2017), 79-88 or ref. 51) which decreases originality of the article. According to in-vitro drug release, cytotoxicity and transport experimental results, the authors have successfully developed gel formulations for nasal administration of clonazepam with high efficiency in terms of permeation, drug solubility and low toxicity. The authors’ experimental results support that RAMEB cyclodextrin has an essential influence on the improved drug delivery properties of their designed gel formulations. As the authors claim in the introduction, although thermosensitive nasal gel benzodiazepines formulations have already been described in previous papers (see ref. 36), this four-component gel formulation is a novelty. Probably, inclusion of the cyclodextrin derivative in the formulation is the main novelty although cyclodextrins have extensively been used in many drug delivery systems (i.e.: Lachowicz Drug Targets (2020), 21, 1495-1510).
I found the manuscript well-written with few typing mistakes and hardly any format incogruence. However, I found acronym use in the article a bit mixed up, wherefore I suggest the authors should correct or clarify it. The preformulation, preparation and characterization of drug-loaded gel formulations, and cytotoxicity and transport studies are concisely and clearly described. The results and discussion clearly support the author’s conclusions. I only regret that the authors do not include any in-vivo experiment about the effectiveness of their formulation. The abstract and keywords provided by the authors are meaningful and accurately represent the article.
All in all, the manuscript contents fulfil the scientific standards of Pharmaceutics.
Specific comments:
- Line 12, CLZ acronym firstly appears in text without the corresponding name of the drug: clonazepam.
- Line 34, I think CLZ should be followed by “(clonazepam)”.
- Lines 304-305, references about this "subject-matter knowledge" should be included if these preliminary experiments have previously been published. If not, experiments should be described.
- Line 364, authors have used the notation “PLX*RAMEB” in the text, whereas in Figure 2 (line 353) they have used “PLX*Cd”. They have used these two terms to mention the same compound, the ramdomized methylated-beta-cyclodextrin, in the text, in figures and tables (see lines 369, 381, 386, 393, 403, 421, 422, etc.., Table 1, 401). I would recommend them to use the same acronym throughout the article in order to ease the reader’s comprehension (i.e: use always Cd).
- Figure 6, in abscissa axis I would recommend indicating the meaning of numbers. Also in the legend “Gel G5 w/o RAMEB” is confusing. I think that using “Gel G5 without Cd” is clearer.
- Line 511. I would type “LY (yellow lucifer)” and “BL (basolateral)” in order to make reading of the article easier.
- References 3, 5, 10, 13, 17, 27, 29, 35, 44, 49, 50 and 53 all the words of titles begin with a capital letter. To homogenize the format of titles in references, only the first word of the title should start with a capital letter.
- Line 572, “Clin Pharmacokinet.” should be typed “Clin. Pharmacokinet.”
- Line 582, “J. Pharm. Pharm. Sci.” should be typed “J. Pharm. Sci.”
- Line 587, “Pharm Nanotechnol.” should be typed “Pharm. Nanotechnol.”
- Lines 614 and 616, “Expert Opin Drug Deliv.” should be typed “Expert Opin. Drug Deliv.”
- Line 623, “Drug Dev Res.” should be typed “Drug Dev. Res.”
- Line 670, “Indo Am. J. Pharm.” should be typed “Indo Am. J. Pharm. Res.”

Author Response
Thank you very much for your comments. Statistical design of experiments (DoE) is an approach that can be applied to several applications in different fields. In this sense, I don't think that the use of such a versatile technique can reduce the originality of the work. Concerning the in-vivo experiment, I'm agree with the Reviewer, the experiments are work in progress: we are currently awaiting a feasibility assessment by the Ethics Committee of the University of Florence. All the changes in the manuscript were highlighted.
Specific comments:
Q1-Line 12, CLZ acronym firstly appears in text without the corresponding name of the drug: clonazepam.
A1-The name of the drug was added
Q2-Line 34, I think CLZ should be followed by “(clonazepam)”.
A2-The name of the drug was added
Q3-Lines 304-305, references about this "subject-matter knowledge" should be included if these preliminary experiments have previously been published. If not, experiments should be described.
A3-The references were added
Q4-Line 364, authors have used the notation “PLX*RAMEB” in the text, whereas in Figure 2 (line 353) they have used “PLX*Cd”. They have used these two terms to mention the same compound, the ramdomized methylated-beta-cyclodextrin, in the text, in figures and tables (see lines 369, 381, 386, 393, 403, 421, 422, etc.., Table 1, 401). I would recommend them to use the same acronym throughout the article in order to ease the reader’s comprehension (i.e: use always Cd).
A4-Thank you for the comment. Actually, we used these two terms, Cd and RAMEB, as synonymous, considering that RAMEB was the only Cd used in this study. However, to avoid possible misunderstanding, according to the Reviewer’s suggestion, now we used the same acronym throughout the text (including Figures and Tablets). We preferred the use of the term RAMEB, considering it more specific than the generic Cd.
Q5-Figure 6, in abscissa axis I would recommend indicating the meaning of numbers. Also in the legend “Gel G5 w/o RAMEB” is confusing. I think that using “Gel G5 without Cd” is clearer.
A5-The legend has been changed according to the Reviewer’s suggestion, and the meaning of the numbers in abscissa axis has been explained in the legend below
Q6-Line 511. I would type “LY (yellow lucifer)” and “BL (basolateral)” in order to make reading of the article easier.
A7-“Lucifer Yellow” and “basolateral” were added before LY and BL, respectively, at lines 549 and 550p, 15
Q7-References 3, 5, 10, 13, 17, 27, 29, 35, 44, 49, 50 and 53 all the words of titles begin with a capital letter. To homogenize the format of titles in references, only the first word of the title should start with a capital letter.
A7-The references have been corrected
Q8_Line 572, “Clin Pharmacokinet.” should be typed “Clin. Pharmacokinet.”
A8_The reference has been corrected
Q9_Line 582, “J. Pharm. Pharm. Sci.” should be typed “J. Pharm. Sci.”
A9_The citation is correct: Pires A, Fortuna A, Alves G, Falcão A. Intranasal drug delivery: how, why and what for? J Pharm Pharm Sci. 2009;12(3):288-311. doi: 10.18433/j3nc79. PMID: 20067706. The journal is Journal of Pharmacy and Pharmaceutical Sciences
Q10_Line 587, “Pharm Nanotechnol.” should be typed “Pharm. Nanotechnol.”
A10_The reference has been corrected
Q11_Lines 614 and 616, “Expert Opin Drug Deliv.” should be typed “Expert Opin. Drug Deliv.”
A11_The reference has been corrected
Q12_Line 623, “Drug Dev Res.” should be typed “Drug Dev. Res.”
A12_The reference has been corrected
Q13_Line 670, “Indo Am. J. Pharm.” should be typed “Indo Am. J. Pharm. Res.”
A_13_The reference has been corrected
Reviewer 2 Report
The manuscript is described a drug release gel materials using cyclodextrins. The addition of RAMEB improved the cell viability and permeability in addition to the improvement in drug retention and release. The authors fully investigate the conditions of gel formation and biochemical experiments such as cytotoxicity and permeability coefficient. However, the effect of adding RAMEB is interesting, but the mechanism is unknown. Thus, after the authors have addressed the comments below, this work would be suitable for publication in Pharmaceuticals.
- The authors should show chemical structure of the key compounds, such as CLZ, PLX, RAMEB, Chito Gl, and HA.
- The mechanism of gelation should be illustrated for ease of understanding.
- It is difficult to understand difference between Cd and RAMEB, for example in Table 1 and Fig. 2, etc. If the meaning of Cd is RAMEB, all Cd should be rewritten to RAMEB.
- RAMEB is known to include the PPO part. Which of the PPO and CLZ is included in RAMEB when CLZ is added?
- 5 should show the percentage of CLZ released from the amount of CLZ incorporated into the gel, in addition to the amount of CLZ released. At that time, if there is a difference in speed depending on the amount of RAMEB added, why?
Author Response
Thank you very much for your comments Please find point-by-point response to the your’s comments . All the changes in the manuscript were highlighted.
Q1_The authors should show chemical structure of the key compounds, such as CLZ, PLX, RAMEB, Chito Gl, and HA.
A1_Chemical structures of CLZ, PLX, RAMEB, ChitoGl and HA were added as supplementary material
Q2_The mechanism of gelation should be illustrated for ease of understanding.
A2_The gelation mechanism was better explained at lines 77-84
Q3_It is difficult to understand difference between Cd and RAMEB, for example in Table 1 and Fig. 2, etc. If the meaning of Cd is RAMEB, all Cd should be rewritten to RAMEB.
A3_Thank you for the comment. Actually, we used these two terms, Cd and RAMEB, as synonymous, considering that RAMEB was the only Cd used in this study. However, to avoid possible misunderstanding, according to the Reviewer’s suggestion, now we used only the term RAMEB throughout the text (including Figures and Tablets).
Q4_RAMEB is known to include the PPO part. Which of the PPO and CLZ is included in RAMEB when CLZ is added?
A4_Since the interaction between cyclodextrin and lipophilic compounds is a reversible process is it possible that an equilibrium between the inclusion of drug and other lipophilic components occurs.
Q5_ should show the percentage of CLZ released from the amount of CLZ incorporated into the gel, in addition to the amount of CLZ released. At that time, if there is a difference in speed depending on the amount of RAMEB added, why?
A5_As now more clearly explained in the text (lines 496-502 page 13) , what we meant to say is that “the amount of drug released as a function of time increased with increasing the CLZ content in the formulation”; that is the increase in the amount of drug released passing from G1 to G3 to G5 gels is due to the corresponding increase of the drug content in these formulations (and not to the increase in the RAMEB amount).
However, as described in the text in the following lines, “a clear favourable effect of RAMEB in improving the drug release rate was observed. In fact, the amount of drug released at each time from G1, G3 and G5 formulations was always significantly higher than that from the corresponding formulations without RAMEB”. As now explained, this result could be reasonably attributed to the solubilizing effect of RAMEB towards CLZ [see ref. 31].
In other words, at equal concentration of drug, the presence of RAMEB allowed to increase the drug release rate, in comparison with the same formulations, containing the same amounts of drug, but without RAMEB (compare G1 with RAMEB with G1 witout RAMEB, and so on…).
Round 2
Reviewer 2 Report
My comments were adequately addressed.